# Job satisfaction among healthcare workers in the aftermath of the COVID-19 pandemic

**Emilia Barili[1] , Paola Bertoli[2] , Veronica Grembi [3] * , Veronica Rattini[4]**

**1** Department of Economics, University of Genova, Genova, Italy, **2** Department of Economics, University of Verona, Verona, Italy, **3** Department of Social Sciences and Economics, Sapienza University of Rome, Rome, Italy, **4** Department of Economics, Management and Quantitative Methods, University of Milano, Milano, Italy

☯ These authors contributed equally to this work.
* veronica.grembi@uniroma1.it

**Data Availability Statement:** The data underlying the results presented in the study are available from Harvard Dataverse (10.7910/DVN/KFOQPD).

## Abstract

Using a unique survey of more than 7,000 respondents conducted immediately after the first wave of the COVID-19 pandemic in Italy, we investigate potential drivers of the job satisfaction of healthcare workers. Relying on a representative sample of Italian physicians and nurses, we show that, in addition to personal characteristics (e.g., age, gender, health status), contextual factors (i.e., working conditions) play the leading role in explaining variation in the level of satisfaction (58%). In particular, working in a high-quality facility increases worker satisfaction and willingness to remain in the profession, and in the current medical specialization, while working in a province with a perceived shortage of medical personnel yields the opposite result. Direct experience with COVID-19 (e.g., having tested positive) is not significantly correlated with the level of job satisfaction, which is instead significantly reduced by changes in the working conditions caused by the health emergency.

## Introduction

The job satisfaction of healthcare personnel is a crucial factor in healthcare management, as it has been found to be directly linked to higher quality of care, greater patient adherence to treatments, and higher patient satisfaction [1, 2]. The recent pandemic has escalated the problem of low job satisfaction among healthcare workers, further threatening the sustainability of healthcare systems. The US [3, 4] and the UK [5] are experiencing a staffing crisis due to workers quitting or retiring early, exhausted by the health emergency, and soon other countries might also have to address the consequences of an exodus of healthcare workers triggered by the pandemic. For instance, according to a survey of the Italian Association of Executive Physicians [6], nearly half of the physicians currently working for the National Health System wish to quit their position in the next two years. Similarly, 4% of Spanish doctors report their intention to leave the profession, while 30% admit considering this option [7]. These challenges, combined with the shortages of healthcare workers already experienced by some countries, might further compromise the quality and safety of patient care. Indeed, the prepandemic estimates of the European Commission indicate a gap in the supply of healthcare human resources of approximately one million workers in 2020, meaning that nearly 15% of the health

**Funding:** Paola Bertoli is the recipient of a Rita Levi Montalcini Fellowship to promote the moving back in Italian University of Young Italian Scholars based abroad and willing to come back to Italy. The funder had no role in study design, data collection and analysis, decision to publish, or preparation of the manuscript.

**Competing interests:** The authors have declared that no competing interests exist.

needs of the EU population was not adequately covered [8]. Recent updates estimate the EU shortage of health workers to increase to approximately 4.1 million units by 2030: 0.6 million physicians, 2.3 million nurses and 1.3 million other healthcare professionals [9]. Therefore, exploring the determinants of healthcare workers' professional satisfaction and vocation in the aftermath of the pandemic becomes crucial to defining the areas of intervention to support the sustainability of healthcare systems in the long run.

We addressed this issue using a unique 50-question survey with 7,681 respondents conducted immediately after the first wave of the COVID-19 outbreak (February-May 2020) in Italy, one of the countries most affected by the pandemic. Among EU countries, Italy was the first to register more than 20,000 deaths, and it reached this threshold between the end of January 2020 and April 14, 2020. Italy was also the country with the second-highest number of deaths (120,053 compared to 128,136 in the UK) by the end of April 2021, approximately a year after the start of the pandemic [10]. During the first wave, the COVID-19 mortality rate and contagiousness were extremely heterogeneous by region, and the outbreak was more severe in the North of the country, with remarkable regional variation (S1 Fig). To control the rapid spread of the outbreak, uniform measures were taken at the national level (e.g., case-detection, contact-tracing, isolation, physical distancing). However, each Italian region was responsible for the actual implementation of these interventions within its territory, and the response to the pandemic ultimately differed substantially both in means and timing across the country. In any case, enormous efforts were made to reorganize the available healthcare resources. These include the reallocation of health personnel from ordinary wards to the treatment of COVID-19 patients, the recruitment of additional health personnel, the increase in the number of intensive care units (ICUs) and beds by converting ordinary hospital wards to ICUs and creating temporary hospitals. Overall, during the first wave, healthcare workers faced an unprecedented situation, and their work was undermined by continuous changes in the health procedures and by frequent shortages of protective equipment, which increased their risk of infection. Unlike the general public, they were excluded from the preventive quarantine measures prescribed after having a contact with COVID-19 positives, and they could stop working only in the event of experiencing respiratory symptoms or if they tested positive. Finally, their greater risk of contagion was disregarded for most of the first wave.

Our interest lies in understanding the main channels driving the overall level of job satisfaction among healthcare workers in the aftermath of the pandemic. We proxied for job satisfaction with direct questions about it, as well as with questions on the respondents' willingness to leave the profession or to move to another specialization. This type of analysis is important because, while the literature on the consequences of the pandemic has addressed the mental and psychological hardships suffered by the healthcare workforce due to the COVID-19 emergency, [11–19] there is scant evidence on the job satisfaction experienced [20–23]. Moreover, most of this evidence relies on small or selected samples, such as professionals (mostly nurses only) working in a specific hospital/region. Differently, our sample is comparable to the full population of workers, both in terms of the age distribution and gender composition, which are determinant factors in explaining job satisfaction.

Finally, to study the channels driving the job satisfaction of healthcare workers, we focused on both groups of drivers identified by the literature [24–28]: a traditional group of drivers, consisting of personal and contextual factors (e.g., age, wages, and workload) and a COVID-19 group of drivers (e.g., being exposed to the virus). Our survey collected information on both groups. Personal and contextual factors included socioeconomic measures and the characteristics of the workplace (e.g. type of hospital, type of employment contract). The COVID-19 controls included questions on personal experience with the pandemic, such as testing positive for the virus, working with COVID-19 patients, working overtime due to the health emergency,

having infected colleagues, or losing colleagues due to the virus. Finally, we also considered administrative data on the COVID-19 first wave mortality rate in the province of work (108 provinces) as an out-of-survey robustness measure.

## Methods

We conducted an online anonymous survey using the Google Form platform, including 50 short questions—translation available in the companion paper [17]. As described by S2 Fig, answers were collected between June 15 and August 31, 2020. Potential participants received an initial invitation email, followed by two reminders, one and two weeks after the first invitation (S3 Fig). The invitation email explained that participation was possible through the use of any electronic device (i.e., PC, tablet, or smartphone) and an internet connection. Potential participants were also informed that the expected completion time was approximately 15 minutes.

### Participants

We primarily contacted potential participants through individual email addresses, having recovered their contact information from various sources: provincial boards of physicians and nurses (108 provinces), hospital websites, and representative associations, some of which also agreed to advertise and share our survey on their website, as reported in S1 Table. Overall, we collected 7,681 answers distributed among 33.2% (2,549) physicians, 59.4% (4,561) nurses, and 7.4% (571) other health workers (e.g., technicians, biologists, safety inspectors, administrative personnel, and researchers). The inclusion of other health workers in addition to physicians and nurses was important to capture the impact of COVID-19 on these professionals, who were often reassigned as contact-tracers, and to account more accurately for the regional disparities in the availability of healthcare personnel. As shown in S4 Fig, our main focus was on the northern areas since they were the most affected, and at the same time, we encountered a general low response rate of workers form southern areas. Specifically, we had 2,797 nurses, 1,657 physicians and 400 other health workers from the northern regions (i.e., Piedmont, Valle d'Aosta, Lombardy, Trentino-Alto Adige, Veneto, Friuli-Venezia Giulia, Liguria, and Emilia-Romagna). We had 1,999 respondents (i.e., 1,206 nurses, 685 physicians and 108 other health workers) from central regions and 828 respondents (i.e., 558 nurses, 207 physicians and 63 other health workers) from southern regions. Note that it is not possible to compute our exact response rate since our survey also circulated through the provincial boards of physicians and nurses, hospital websites, and representative associations, some of which also agreed to advertise and share our survey on their website. In any case, different from most previous studies, our survey targeted all Italian healthcare workers rather than workers working in specific hospitals or geographical areas within the country. In addition, note that having a difference in survey responses across Italian macro areas is quite common in this type of empirical study [29–31].

Table 1 compares the distribution of our sample in terms of gender, profession, and region of work (Columns 1 and 3) with respect to the administrative data on the 2019 population of physicians and nurses (Columns 2 and 4). It appears that we achieved good representativeness along the gender composition dimension in the most pandemic-affected areas (i.e., Piedmont, Lombardy, Veneto and Emilia-Romagna), both among physicians and nurses. Indeed, the average percentages of females in the North was equal to 50.1% among physicians and almost 80% among nurses, which are very much in line with the national averages (i.e., 50.8% and 84.5%, respectively). Regarding age composition, our sample was slightly younger as

**Table 1. Sample composition and national population statistics.**

| Region | %Women Physicians Sample | %Women Physicians National Level | %Women Nurses Sample | %Women Nurses National Level |
|---|---|---|---|---|
| **Northern Regions** | | | | |
| Piedmont | 47.7 | 50.9 | 77.3 | 84.3 |
| Valle d'Aosta | 33.3 | 47.5 | 100.0 | 89.4 |
| Lombardy | 53.4 | 52.3 | 77.4 | 83.0 |
| Prov. Autonome Trento-Bolzano | 59.4 | 48.5 | 74.9 | 87.1 |
| Veneto | 53.8 | 48.9 | 73.1 | 83.1 |
| Friuli Venezia Giulia | 37.5 | 53.1 | 80.0 | 85.0 |
| Liguria | 61.5 | 50.2 | 75.8 | 82.5 |
| Emilia Romagna | 54.0 | 54.7 | 74.3 | 81.3 |
| **Central Regions** | | | | |
| Tuscany | 49.9 | 51.8 | 72.5 | 81.3 |
| Umbria | 59.3 | 50.1 | 63.2 | 77.5 |
| Marche | 55.5 | 50.4 | 74.1 | 80.8 |
| Lazio | 45.7 | 47.7 | 68.9 | 76.4 |
| **Southern Regions and Islands** | | | | |
| bruzzo | 37.0 | 49.6 | 66.9 | 77.8 |
| Molise | 75.0 | 40.8 | 46.2 | 76.9 |
| Campania | 28.6 | 36.2 | 53.9 | 62.0 |
| Puglia | 33.3 | 43.0 | 54.2 | 71.4 |
| Basilicata | 25.0 | 38.9 | 42.9 | 73.9 |
| Calabria | 23.1 | 40.1 | 45.8 | 64.3 |
| Sicily | 24.6 | 40.9 | 48.0 | 60.4 |
| Sardinia | 38.2 | 57.5 | 67.7 | 81.0 |
| Average | 44.8 | 47.6 | 66.8 | 77.9 |
| Average (North only) | 50.1 | 50.5 | 79.1 | 84.7 |

physicians on average were 49 years old and nurses 40, while at the national level, the two groups recorded an average age of 52 and 47 in 2018 (the last available year) [32].

## Ethics

The study protocol was approved by the "Comitato di Approvazione per la Ricerca sull'Uomo", that is, the Ethics Committee of the University of Verona (Prot. N. 0221872—22/06/2020). The protocol was also registered at the AEA-RCT registry (AEARCTR-0007419), while the University of Pavia certified compliance with privacy requirements (Prot. N. 61080—15/06/2020). All participants gave their written informed consent that was embedded on the first page of the questionnaire. After reading a description of the questionnaire, healthcare professionals were asked if they agreed to participate. If they ticked the option "proceed" after the statement "I confirm that I have read the information on the processing of personal data and I agree to participate in this survey" on the electronic form, the survey would begin. Respondents voluntarily participated and could withdraw from the survey at any time.

## Outcomes and covariates

The outcome of the study is job satisfaction. This is directly captured by $Satisfaction_i$ and indirectly proxied by the willingness to change jobs or medical specialization. As described in Eq 1, $Satisfaction_i$ is an index that varies from 0 to 8 by summing up 8 dummies ($D_{ci}$) referring to

the following aspects: *Profession*, *Job*, *Salary*, *Work-life balance*, *Relationships with colleagues*, *Relationships with the administration*, *Work hours*, and *Career*. Since the aspects along which satisfaction is evaluated are originally measured on a 5-item Likert scale, each related dummy takes value 1 if the respondent stated being satisfied or very satisfied with the aspect recalled by the name of the dummy itself. Note that our measure of job satisfaction is constructed using the questions from the Labor Force Survey ("Rilevazioni Forza Lavoro—RFL") by the National Institute of Statistics (Istat). For a detailed definition of the outcome variables and the related dummies, see S2 Table.

$$Satisfaction_i = \sum_{c=1}^{8} D_{ci} \quad \text{where} \quad D_{ci} = \begin{cases} 1 & \text{satisfied or very satisfied with dimension } c \\ 0 & \text{otherwise} \end{cases} \tag{1}$$

The variable *Profession Change* (*Specialization Change*) is instead defined by the answer given to a unique statement: "If I could start over, I would not be in this profession" ("If I could start over, I would choose a different field of specialization"). Then, it is a dummy equal to 1 if the respondent agreed or strongly agreed with the statement (the level of agreement was originally measured on a 5-item Likert scale). Alternative ways to construct the outcomes of interest are discussed in Section.

Table 2 reports the cross-correlations between the outcomes. As expected, *Satisfaction* is strongly and negatively correlated with *Profession Change* and *Specialization Change*, which are in turn positively correlated with one another. The distribution of the outcome variables

**Table 2. Cross-correlations between outcomes of interest, alternative outcomes and single components.**

| | Satisf | Prof Change | Spec Change | Satisf 2 | Satisf 3 | Satisf PCA | Satisf: Prof | Satisf: Job | Satisf: Salary | Satisf: Balance | Satisf: Colleagues | Satisf: Admin | Satisf: Work hours | Satisf: Career |
|---|---|---|---|---|---|---|---|---|---|---|---|---|---|---|
| **Outcomes of interest:** | | | | | | | | | | | | | | |
| Satisfaction | 1.000 | | | | | | | | | | | | | |
| Profession Change | -0.321 | 1.000 | | | | | | | | | | | | |
| Specialization Change | -0.281 | 0.483 | 1.000 | | | | | | | | | | | |
| **Alternative outcomes:** | | | | | | | | | | | | | | |
| Satisfaction 2 | 0.909 | -0.369 | -0.328 | 1.000 | | | | | | | | | | |
| Satisfaction 3 | 0.909 | -0.369 | -0.328 | 1.000 | 1.000 | | | | | | | | | |
| Satisfaction PCA | 0.907 | -0.372 | -0.332 | 1.000 | 1.000 | 1.000 | | | | | | | | |
| **Individual components:** | | | | | | | | | | | | | | |
| Satisf: Profession | 0.517 | -0.454 | -0.369 | 0.617 | 0.617 | 0.626 | 1.000 | | | | | | | |
| Satisf: Job | 0.636 | -0.324 | -0.318 | 0.724 | 0.724 | 0.736 | 0.564 | 1.000 | | | | | | |
| Satisf: Salary | 0.603 | -0.226 | -0.179 | 0.641 | 0.641 | 0.624 | 0.340 | 0.384 | 1.000 | | | | | |
| Satisf: Work-life balance | 0.637 | -0.205 | -0.173 | 0.673 | 0.673 | 0.671 | 0.276 | 0.346 | 0.348 | 1.000 | | | | |
| Satisf: Colleagues | 0.559 | -0.181 | -0.179 | 0.634 | 0.634 | 0.627 | 0.276 | 0.415 | 0.223 | 0.305 | 1.000 | | | |
| Satisf: Administration | 0.616 | -0.177 | -0.155 | 0.676 | 0.676 | 0.677 | 0.276 | 0.386 | 0.323 | 0.338 | 0.498 | 1.000 | | |
| Satisf: Work hours | 0.621 | -0.175 | -0.172 | 0.667 | 0.667 | 0.670 | 0.258 | 0.341 | 0.289 | 0.626 | 0.311 | 0.385 | 1.000 | |
| Satisf: Career | 0.666 | -0.257 | -0.233 | 0.720 | 0.720 | 0.720 | 0.385 | 0.462 | 0.466 | 0.356 | 0.359 | 0.417 | 0.373 | 1.000 |

Correlation coefficients between outcomes of interest, alternative outcomes and single components. See S2 Table for the variable definition.

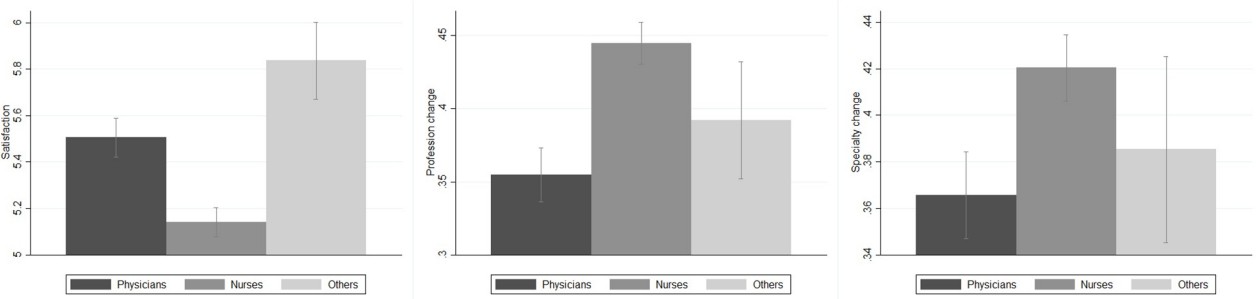

**Fig 1. Outcomes of interest among professions.** *Others* identifies healthcare workers other than physicians and nurses, i.e., safety inspectors, controllers, administrative personnel, biologists, and researchers. *Satisfaction* (A) is a measure taking values between 0 and 8, with 8 representing the highest level of satisfaction. *Profession change* (B) and *Specialization change* (C) are two dummies taking value 1 if the respondent reported a high propensity to change profession and medical specialization, respectively. For a detailed description of these variables, see S2 Table.

within professions is shown in Fig 1. Physicians experience a level of satisfaction that is higher than that of nurses but lower than that of other healthcare professionals. Consistently, nurses are more prone to change both profession and medical specialization than physicians, while other professionals place themselves in between nurses and physicians, although the confidence interval (95%) of the values across professions overlaps. This is consistent with the expectation that since the training for a physician is significantly longer than that for a nurse or for a lab technician, physicians have a higher cost of switching to a different profession.

Consistent with the literature [20], we grouped the covariates into personal, contextual, and COVID-19-related factors as summarized in Table 3.

**Personal factors.** The first group of covariates included the socioeconomic characteristics and basic attributes of healthcare workers, from their gender and health conditions to a proxy for their household wealth as the dimension of their home. Several questions were included to capture the possible higher distress due to the fear of infecting others or having relatives who could become infected (i.e., home dimension, living alone, having health workers in the

**Table 3. Covariates.**

| Personal factors | Contextual factors | COVID-19-related factors |
|---|---|---|
| Children | Hospital worker | COVID-19 Death rate |
| Age | Teaching hospital | Prompt response |
| Female | Private | Effective response |
| Italian | Managerial role | Infected colleagues |
| Married | Contract with work-shifts | Dead colleagues |
| Home sq. meter > 100 | Average hours worked | COVID-19 overtime |
| Good health | Tenure | Exposed to COVID-19 |
| Living alone | COVID-19 related specialization | Positive to COVID-19 |
| Never changed workplace | High-quality facility | Work with COVID-19 positives |
| Health workers in the family | Lack of medical personnel | COVID-19: change of specialization/function |
| Chronic diseases | High salary | |
| | Nurse | |

When we refer to the COVID-19 crisis, we refer to the first wave that took place in Italy from the end of February 2020 to the beginning of June 2020

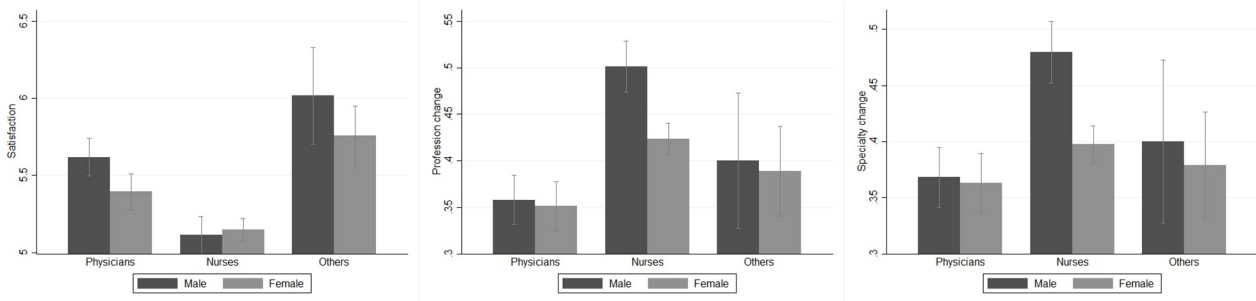

**Fig 2. Outcomes of interest among professions and by gender.** *Others* identifies healthcare workers other than physicians and nurses, i.e., safety inspectors, controllers, administrative personnel, biologists, and researchers. *Satisfaction* (A) is a measure taking values between 0 and 8, with 8 representing the highest level of satisfaction. *Profession change* (B) and *Specialization change* (C) are two dummies taking value 1 if the respondent reported a high propensity to change profession and medical specialization, respectively. *Gender* is a dummy taking value 1 if the respondent is female, 0 if male. For a detailed description of these variables, see S2 and S3 Tables.

family). As shown in Fig 2, female and male nurses almost do not differ along *Satisfaction*, while female physicians are more generally unsatisfied with their working conditions than their male colleagues. Differences also persist in the willingness to change profession and specialization. While physicians are less willing to change profession or specialization regardless of their gender than nurses, male nurses are more willing to change than female nurses.

We defined *ad hoc* variables that may capture nuanced differences in the outcomes: the presence of healthcare workers in the family of origin and whether the respondent has always been employed in the facility where she is employed at the time of the survey. Regarding the professional background of the family of origin, the effect may be twofold: on the one hand, sharing the same profession and challenges is a source of support in coping with similar problems; on the other hand, the experience of relatives could serve as a benchmark to evaluate one's own working conditions. Having changed workplaces proxies for how well the respondent knows her working environment but also indicates the variety of experience she has in terms of different working environments. This could have a positive or a negative impact on job satisfaction. The square footage of the accommodation provides valuable information since it is an indirect measure of wealth that is not necessarily captured by workers' income (which we control for): an individual earning a low salary could still belong to a wealthy family. We also asked whether survey participants were living alone to account, for example, for unmarried cohabiting couples or non-cohabiting workers.

**Contextual factors.** The second group of covariates controlled for a set of basic characteristics defining the type of worker. Specifically, participants disclose whether they are nurses and hospital workers, work in a public facility and have a managerial/coordinating role since these characteristics make them among the most exposed to the pressure triggered by the health emergency. The top panel of Fig 3 indicates that professionals with managerial responsibilities tend to report a higher level of satisfaction and a lower willingness to change profession than those with no similar responsibilities. Physicians with managerial duties are additionally less willing to change specialization than physicians without managerial duties.

Respondents also provided information on their working conditions because these can impact their satisfaction. These conditions include average working hours, number of years of employment, working in a COVID-19-related specialization (i.e, ICU, anesthesiology, emergency care, cardiology, pulmonary diseases, and infectious diseases), contract with work shifts and monthly salary. From the declared monthly salary, we created a dummy that takes value 1 when the monthly salary is above the median of the distribution in our sample (i.e., above

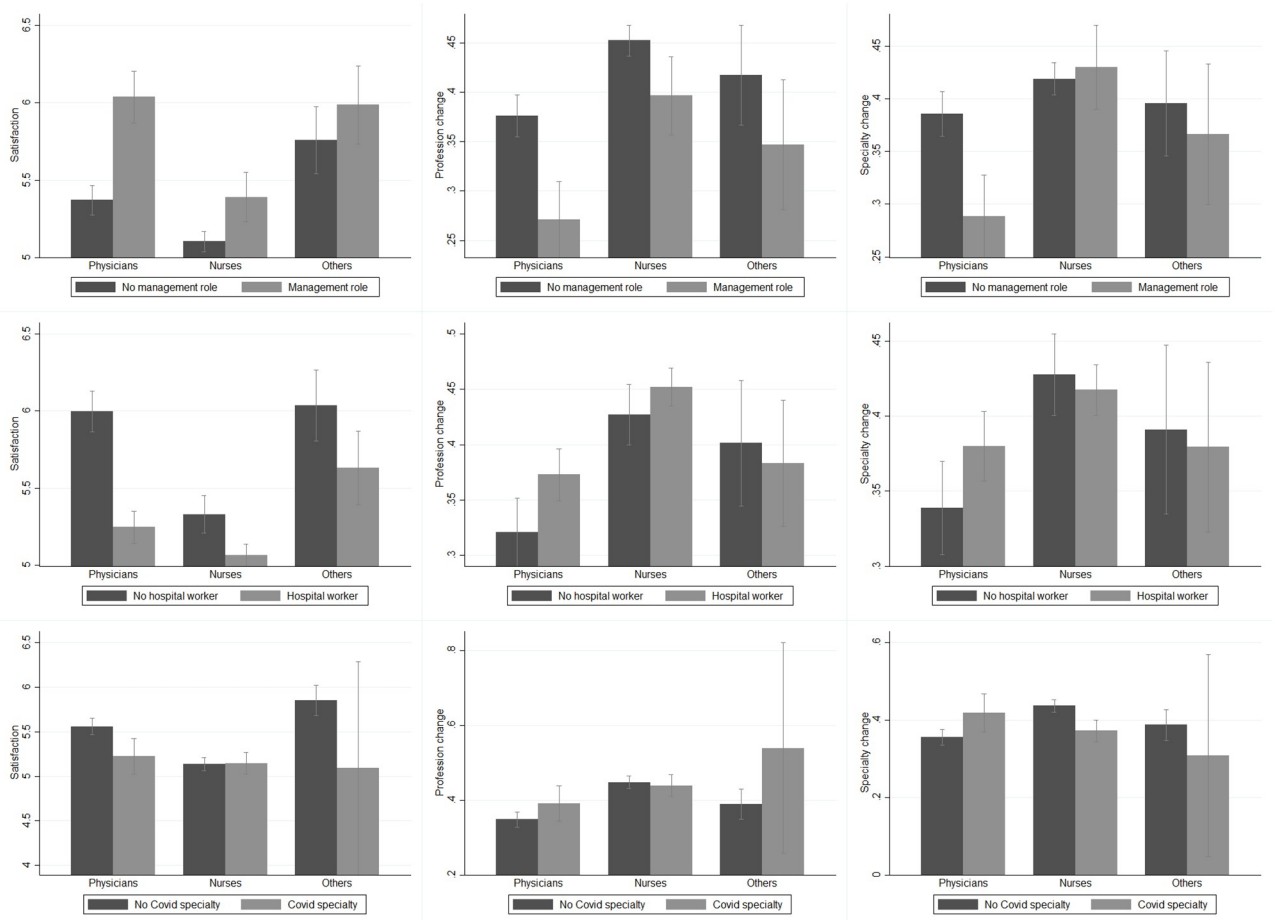

**Fig 3. Outcomes of interest among professions, by role, type of workplace and specialization.** *Others* identifies healthcare workers other than physicians and nurses, i.e., safety inspectors, controllers, administrative personnel, biologists, and researchers. *Satisfaction* (A) is a measure taking values between 0 and 8, with 8 representing the highest level of satisfaction. *Profession change* (B) and *Specialization change* (C) are two dummies taking value 1 if the respondent reported a high propensity to change profession and medical specialization, respectively. *Role* is a dummy taking value 1 if the respondent has a managerial role, 0 otherwise. *Workplace* is a dummy taking value 1 if the respondent is a hospital worker, 0 otherwise. *COVID-19 specialization* refers to COVID-19-related medical specializations, that is, ICU, anesthesiology, emergency care, cardiology, pulmonary diseases and infectious diseases; *No COVID-19 specialization* refers to all other medical specializations. For a detailed description of these variables, see S2 and S3 Tables.

2,000 euros per month). Within professions, hospital workers are less satisfied than their colleagues working outside hospitals, with physicians being more willing to change profession if working in a hospital (central panel of Fig 3). Surprisingly, working in a COVID-19-related specialization is not associated with any specific direction of satisfaction or the professionals' attitudes toward their profession/specialization (bottom panel of Fig 3). If anything, physicians working in COVID-19-related specializations report a lower satisfaction and a higher willingness to change specialization.

Finally, we also considered the quality of the work environment because this is likely to affect job satisfaction. To this end, respondents were asked to report whether they work for a teaching hospital (i.e., a high-quality hospital) and to assess the perceived lack of medical personnel in their province of work and the quality of their employing facilities. The former is a dummy that takes value 1 if the respondent judges that there is a severe or a very severe lack of healthcare personnel in her province of work that might compromise patients' access to care.

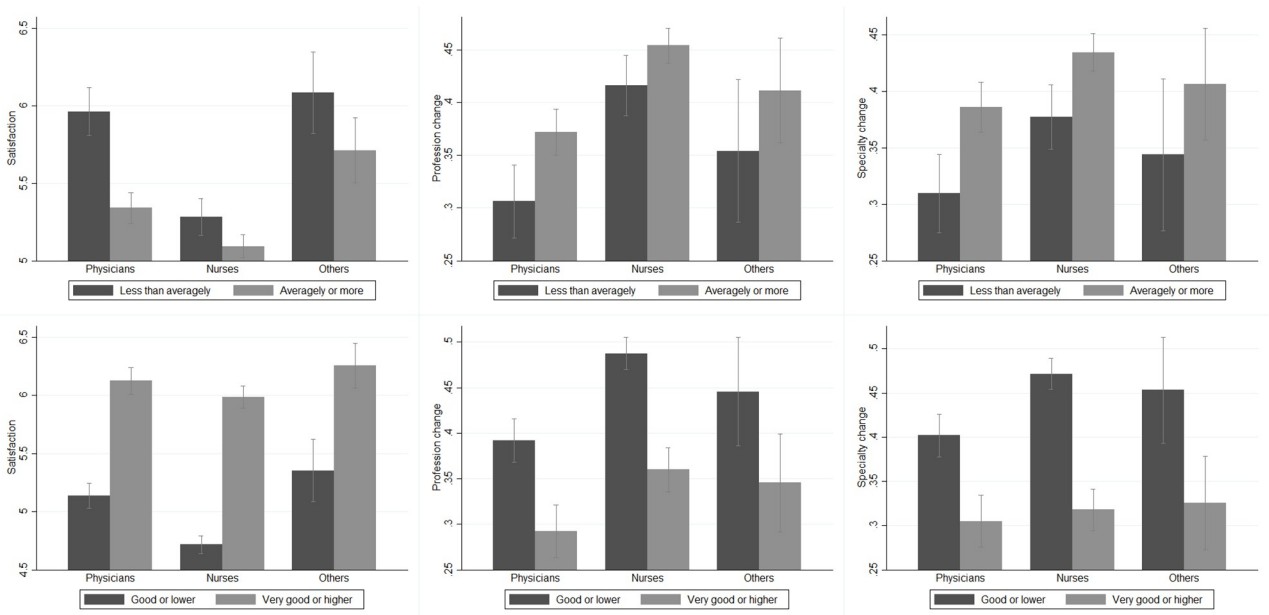

**Fig 4. Outcomes of interest among professions, by lack of personnel and quality.** *Others* identifies healthcare workers other than physicians and nurses, i.e., safety inspectors, controllers, administrative personnel, biologists, and researchers. *Satisfaction* (A) is a measure taking values between 0 and 8, with 8 representing the highest level of satisfaction. *Profession change* (B) and *Specialization change* (C) are two dummies taking value 1 if the respondent reported a high propensity to change profession and medical specialization, respectively. *Lack of personnel* is a dummy equal 1 for a medium to high lack of the medical personnel in the province of work, 0 otherwise. *Quality* is a dummy equal to 1 for workplaces with very good or higher quality, 0 otherwise. For a detailed description of these variables, see S2 and S3 Tables.

The latter is again a binary variable equal to 1 if the respondent defines the facility she works for as being of very good or excellent quality. As shown in the top panel of Fig 4, health workers who perceived a lack in medical personnel tend to report a lower satisfaction and a higher propensity to declare a profession or specialization change, with physicians driving the effect. Conversely, the bottom panel of Fig 4 graphically describes how a perceived higher quality of the facility is associated with a higher level of satisfaction and a lower propensity to change profession or specialization. The effect is large and significant across all professions.

**COVID-19 related factors.** To account for the links between the COVID-19 pandemic and the level of job satisfaction, the third and last group of covariates includes both administrative data and the personal experience with the pandemic. As administrative measure of the COVID-19 outbreak, we relied on the COVID-19 mortality rate as computed by the National Institute of Statistics (Istat) together with the Istituto Superiore di Sanitá (Iss) on administrative data [33]. This index, referring to the period January-May 2020, represents the mortality rate due to COVID-19, standardized by the demographic characteristics of the resident population in each province (values expressed per 100,000 inhabitants). As apparent from Fig 5, the administrative mortality rate at the provincial level is associated with a small and not statistically significant difference in any of our outcomes of interest among both physicians and nurses exposed to different intensities of this measure (i.e., high/low mortality rate). These unexpected results can be interpreted as a first signal that the spread of the pandemic per se might not be significantly correlated with the level of commitment of healthcare workers.

As measures of personal COVID-19 experiences in the workplace, we asked respondents to judge the promptness and effectiveness of the policy response to the COVID-19 emergency in the facility where they work. Additionally, we included a set of variables measuring the exposure of the respondents to COVID-19 infection based on their own experience and the

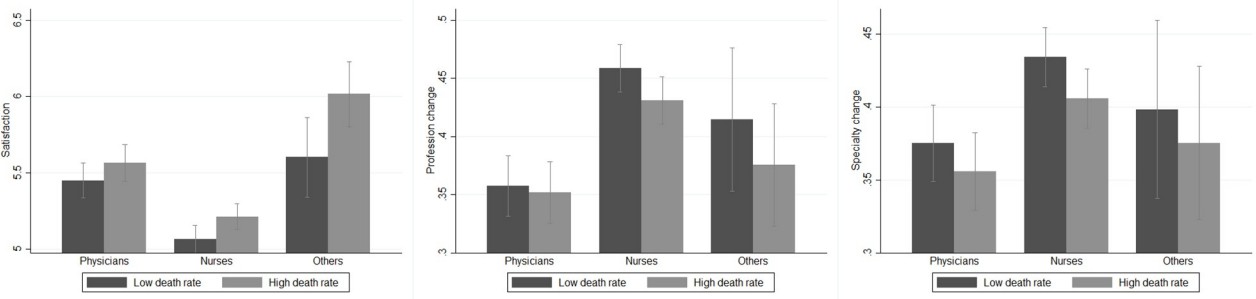

**Fig 5. Outcomes of interest among professions and by covid-19 death rate.** *Others* identifies healthcare workers other than physicians and nurses, i.e., safety inspectors, controllers, administrative personnel, biologists, and researchers. *Satisfaction* (A) is a measure taking values between 0 and 8, with 8 representing the highest level of satisfaction. *Profession change* (B) and *Specialization change* (C) are two dummies taking value 1 if the respondent reported a high propensity to change profession and medical specialization, respectively. *COVID-19 Death rate*, referring to the period January-May 2020, represents the mortality rate due to COVID-19 standardized according to the demographic characteristics of the resident population in each province (values expressed per 100,000 inhabitants); it is a measure computed by the National Institute of Statistics (Istat) together with the Istituto Superiore di Sanità (Iss) on administrative data [33]. For a detailed description of these variables, see S2 and S3 Tables.

experience of their colleagues: whether their colleagues were infected or lost their lives due to COVID-19 and whether respondents themselves were exposed to or tested positive for the disease. Finally, we obtained data on whether respondents directly worked with COVID-19 positives, were reassigned to a specialization or facility devoted to COVID-19 patients, and worked overtime due to the COVID-19 emergency.

As shown in Tables 4–6, nurses are significantly younger and have a higher prevalence of female workers than physicians. Consistently, nurses are less likely to have children, be married, live in large dwellings, cohabit, and suffer from chronic diseases, while they are more likely to have changed workplace before and to not have Italian citizenship. They are also more likely to work in a hospital (especially in a non-teaching hospital) and in the private sector than physicians and to have shorter tenure. However, nurses are less likely to have managerial or coordination roles. Finally, nurses work approximately 38 hours per week compared to the 44 hours recorded among physicians, with fewer work shifts, and they are less likely to judge their unit as being of high quality than physicians. When analyzing the COVID-19-related factors, physicians had a higher chance of having colleagues who were infected/hospitalized or who died of COVID-19, and they also worked more overtime during the first wave than nurses. However, nurses are more likely to have been reassigned to a different specialization or facility and to have tested positive for COVID-19.

## Statistical analysis

For healthcare worker *i* working in region *r*, we estimate the links between each outcome of interest (*Outcome_{ir}*) and the three sets of controls (see Table 3) by applying the model in Eq 2.

$$Outcome_{ir} = \alpha Personal_i + \lambda Contextual_i + \sigma COVID_{19i} + \beta COVID_{19p} + \tau_r + \epsilon_{ir} \quad (2)$$

This model captures the joint impact of workers' personal characteristics (*Personal_i*), contextual conditions (*Contextual_i*), and both personal (*COVID−19_i*) and administrative (*COVID−19_p*, with *p* being the province of work) COVID-19-related factors. In addition, we control for the working region fixed effects $\tau_r$ to account for the time-invariant regional characteristics such as the organization of the regional health system and its performance in regular times, the macro characteristics of the region of work—such as employment or population

**Table 4. Summary statistics: Personal factors.**

|  | All | Physicians | Nurses | P-value: Physicians—Nurses |
|---|---|---|---|---|
| Children | 0.57 | 0.68 | 0.50 | 0.00*** |
|  | (0.50) | (0.47) | (0.50) |  |
| Age | 43.78 | 49.34 | 40.30 | 0.00*** |
|  | (12.69) | (12.29) | (11.81) |  |
| Female | 0.65 | 0.50 | 0.72 | 0.00*** |
|  | (0.48) | (0.50) | (0.45) |  |
| Italian | 0.98 | 0.99 | 0.98 | 0.00*** |
|  | (0.13) | (0.08) | (0.15) |  |
| Married | 0.49 | 0.62 | 0.42 | 0.00*** |
|  | (0.50) | (0.49) | (0.49) |  |
| House sq. meters >100 | 0.51 | 0.68 | 0.41 | 0.00*** |
|  | (0.50) | (0.47) | (0.49) |  |
| Good health | 0.94 | 0.96 | 0.94 | 0.00*** |
|  | (0.23) | (0.19) | (0.24) |  |
| Chronic diseases | 0.36 | 0.38 | 0.35 | 0.02* |
|  | (0.48) | (0.48) | (0.48) |  |
| Living alone | 0.14 | 0.13 | 0.15 | 0.05* |
|  | (0.35) | (0.34) | (0.36) |  |
| Never changed workplace | 0.27 | 0.29 | 0.26 | 0.01** |
|  | (0.44) | (0.45) | (0.44) |  |
| Health workers in the family | 0.34 | 0.33 | 0.35 | 0.07 |
|  | (0.47) | (0.47) | (0.48) |  |
| Obs. | 7681 | 2549 | 4561 | 7110 |

See S3 Table for the variable definitions. Significant at 10% *; significant at 5% **; significant at 1% ***.

characteristics—or the cultural factors that might reflect differences in daily life attitudes and behaviors. Standard errors are clustered at the level of the working region of each respondent $i$.

## Results

Fig 6 shows the share of the explanatory power of personal, contextual and COVID-19-related factors separately as obtained by summing the estimated partial $\eta^2$ for each of the related regressors. We observe a large relative importance of contextual factors with respect to personal and COVID-19-related factors in explaining the outcomes. Contextual factors explain 58% of the variation in *Satisfaction*, 42% of the variation in *Profession Change*, and 52% of the variation in *Specialization Change*. The remaining variation in *Satisfaction* is equally explained by personal (21%) and COVID-19 (21%) related, while the latter only account for a small amount of the variation in *Profession Change* (9%) and *Specialization Change* (9%).

Table 7 shows the regression results for the full sample (i.e., all health workers). We observe a U-shaped reduction in *Satisfaction* with respect to age, with the lowest level of *Satisfaction* appearing among middle-aged respondents (i.e., age 40–50). Workers who are married and in good health show a significantly higher level of satisfaction. Among the contextual factors, working in a high-quality facility is the most important determinant of workers' satisfaction. Those working in a high-quality facility enjoy greater satisfaction by approximately 0.827 percentage points, which is approximately 15.6% at the mean (i.e., 5.3). Perceiving a higher salary

**Table 5. Summary statistics: Contextual factors.**

|  | Full Sample | Physicians | Nurses | P-value: Physicians—Nurses |
|---|---|---|---|---|
| Hospital worker | 0.68 | 0.65 | 0.72 | 0.00*** |
|  | (0.47) | (0.48) | (0.47) |  |
| Teaching hospital | 0.04 | 0.06 | 0.006 | 0.00*** |
|  | (0.18) | (0.23) | (0.18) |  |
| Private | 0.14 | 0.10 | 0.17 | 0.00*** |
|  | (0.35) | (0.30) | (0.35) |  |
| Management role | 0.17 | 0.20 | 0.13 | 0.00*** |
|  | (0.38) | (0.40) | (0.38) |  |
| Contract with work-shifts | 0.74 | 0.69 | 0.83 | 0.00*** |
|  | (0.44) | (0.46) | (0.44) |  |
| Average hours worked | 39.96 | 43.99 | 37.76 | 0.00*** |
|  | (8.45) | (10.18) | (8.45) |  |
| Tenure | 12.41 | 13.44 | 11.53 | 0.00*** |
|  | (11.37) | (11.32) | (11.37) |  |
| COVID-19 related specialization | 0.20 | 0.16 | 0.25 | 0.00*** |
|  | (0.40) | (0.36) | (0.40) |  |
| High-quality facility | 0.36 | 0.37 | 0.33 | 0.00*** |
|  | (0.48) | (0.48) | (0.48) |  |
| Lack of medical personnel | 0.74 | 0.73 | 0.75 | 0.05 |
|  | (0.44) | (0.44) | (0.44) |  |
| High salary | 0.22 | 0.59 | 0.03 | 0.00*** |
|  | (0.42) | (0.49) | (0.42) |  |
| Obs. | 7681 | 2549 | 4561 | 7110 |

See S3 Table for the variable definitions. Significant at 10% *; significant at 5% **; significant at 1% ***.

and having a managerial/coordination role are positively correlated with *Satisfaction*. By contrast, factors reducing workers' satisfaction are the hours of work, having an employment contract with work shifts, and working in a hospital and in a province that is perceived to have a lack of medical personnel. Regarding the impact of COVID-19-related factors, when the response to the crisis was considered to be prompt and effective, workers are overall more satisfied—by approximately 0.6 and 0.4 p.p., respectively (corresponding to a magnitude of 11% and 8% at the mean value). However, those workers who worked more overtime or were reassigned to a different specialization or function are significantly less satisfied.

Column 2 of Table 7 shows the coefficients for the willingness to change profession. In line with previous literature showing a strong correlation between risk aversion, age, and gender [34–36], younger workers and workers with chronic diseases show higher willingness to change profession, whereas female workers, even if overall less satisfied, are less willing to change. In addition, the lack of medical personnel in the province of work, or working more hours, increases the propensity to change profession. In contrast, working in a high-quality facility, receiving a high salary, or having managerial/coordination responsibilities are all employee retention factors. Surprisingly, the first wave of the COVID-19 pandemic did not threaten workers' vocation. In particular, workers who had more contact with COVID-19 patients have a significantly lower willingness to change profession (by 0.03 p.p.). Where there was a more effective response to the emergency, workers are more willing to keep their job in healthcare, while the opposite is true only when workers were reassigned to a different

**Table 6. Summary statistics: COVID-19-related factors.**

|  | All | Physicians | Nurses | P-value: Physicians—Nurses |
|---|---|---|---|---|
| COVID-19 Death rate | 59.76 | 58.35 | 59.88 | 0.32 |
|  | (64.43) | (58.48) | (64.43) |  |
| Prompt response | 0.62 | 0.62 | 0.61 | 0.66 |
|  | (0.49) | (0.49) | (0.49) |  |
| Effective response | 0.79 | 0.80 | 0.78 | 0.03* |
|  | (0.41) | (0.40) | (0.41) |  |
| Infected colleagues | 0.72 | 0.75 | 0.72 | 0.01** |
|  | (0.45) | (0.43) | (0.45) |  |
| Dead colleagues | 0.07 | 0.10 | 0.05 | 0.00*** |
|  | (0.25) | (0.30) | (0.25) |  |
| COVID-19 overtime | 0.68 | 0.79 | 0.62 | 0.00*** |
|  | (0.47) | (0.41) | (0.47) |  |
| Exposed to COVID-19 | 0.09 | 0.09 | 0.10 | 0.02* |
|  | (0.29) | (0.28) | (0.29) |  |
| Positive to COVID-19 | 0.10 | 0.07 | 0.12 | 0.00*** |
|  | (0.30) | (0.26) | (0.30) |  |
| Work with COVID-19 positives | 0.43 | 0.43 | 0.47 | 0.00*** |
|  | (0.50) | (0.49) | (0.50) |  |
| COVID-19: change of specialization/function | 0.21 | 0.17 | 0.24 | 0.00*** |
|  | (0.40) | (0.37) | (0.40) |  |
| Obs. | 7681 | 2549 | 4561 | 7110 |

See S3 Table for the variable definitions. Significant at 10% *; significant at 5% **; significant at 1% ***.

specialization or function due to COVID-19. We obtain very similar results when considering *Change Specialization* (Column 3 of Table 7). Workers employed in COVID-19-related wards or who had been in contact with COVID-19 patients are less willing to change specialization, further signaling that the first wave of the pandemic did not affect their professional vocation.

The disjoint results for physicians (Columns 4, 5, and 6 of Table 7) and nurses (Columns 7, 8, and 9 of Table 7) are consistent with the full sample estimates. Regarding the U-shaped effect driven by age, physicians report the lowest peak in the age group 50–60, while nurses record it among young to middle-aged professionals (i.e., 30–40). The lack of medical personnel (i.e., a lack of human resources or low number of coworkers) negatively impacts the level of satisfaction of physicians, which decreases by 8.8%, but not that of nurses. Managerial or coordination responsibilities also seem to be more important for physicians' satisfaction only. However, both types of professionals are less satisfied with longer working hours and work shifts, while being assigned to a different specialization/function during the first wave of the pandemic decreases satisfaction only for nurses. Regardless of the type of profession, both satisfaction and the willingness to change profession or specialization are driven by the perceived quality of the employing facility. Working in a perceived high-quality facility increases the level of satisfaction by +14.9% and +17.3% for physicians and nurses, respectively, while it decreases the willingness to move to another profession (specialization) by -22.6% (-22.3%) for physicians and by -24.3% (-31.1%) for nurses.

## Robustness checks

We check the robustness of our results along several dimensions. First, to better understand the impact of the COVID-19-specific factors on our outcomes of interest, we include

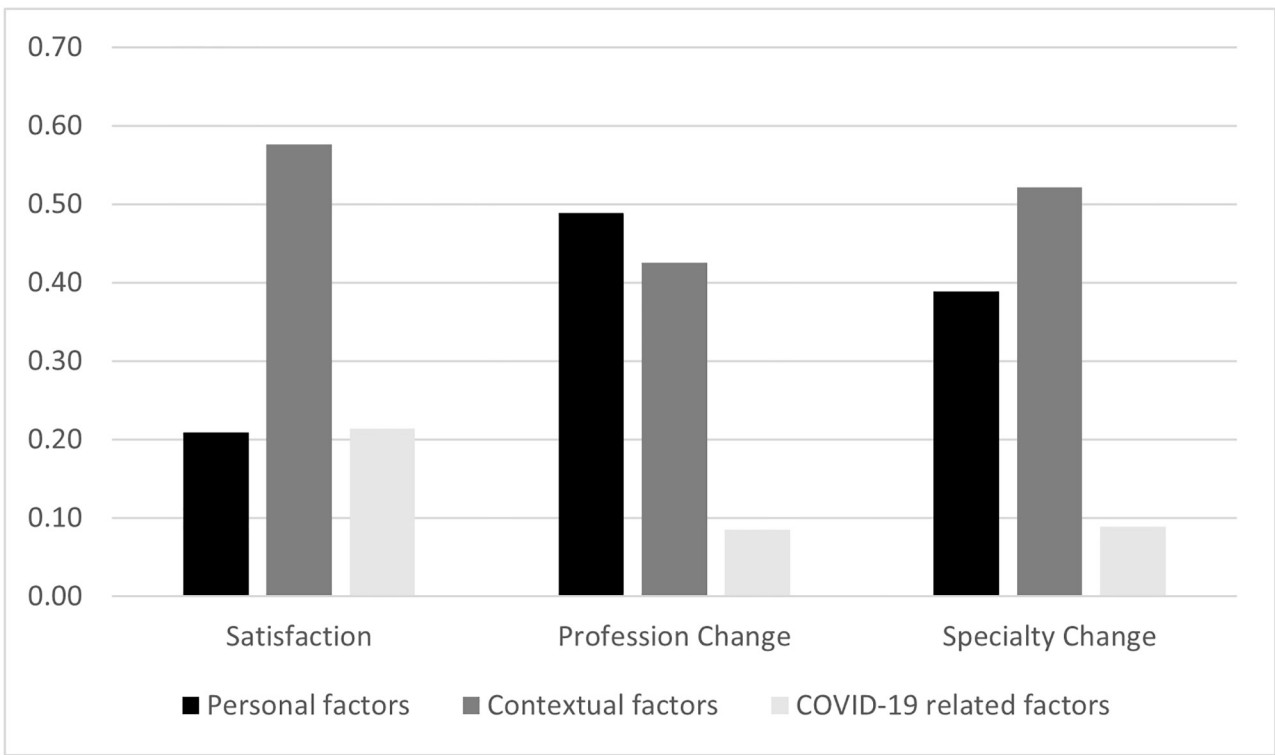

**Fig 6. Post-estimation analysis.** We perform a post-estimation analysis based on the results presented in Table 7. The analysis estimates the partial $\eta^2$ for each regressor included in the model. The figure presents the share of the total explanatory power of the model that can be attributable to the factors within each of the three categories (i.e., personal, contextual, COVID-19-related factors).

separately the administrative variable on the provincial COVID-19 mortality rate and the *COVID-19 factors* derived from the survey data (S4 Table). This robustness check clarifies which type of phenomenon better captures the effects of the COVID-19 pandemic—either the recorded mortality in the province of work or the personal experience of the workers. As is apparent from S4 Table, the inclusion of the COVID-19 mortality rate does not affect the significance of the survey variables describing the workers' personal experience with the pandemic. This suggests that the administrative index and survey data provide different and complementary information. When only the COVID-19 mortality rate is included, we observe a positive effect on *Satisfaction*. As shown in Table 7, in those provinces hit more intensively by the first wave, the level of job satisfaction is higher. This result might be driven by the fact that in the northern provinces, which were more heavily impacted, there is typically a higher level of satisfaction. Moreover, the positive effect of the COVID-19 mortality rate might also reflect both the resilience and the fulfillment that workers experienced during the first wave thanks to emotional support received by the general public. The media often referred to healthcare workers as "heroes," and many public figures, such as Pope Francis, openly thanked them for their heroic services and praised their dedication, while individuals undertook many private initiatives to express their gratitude (e.g., from individual messages to private donations). Overall, this unexpected public reaction might have given further meaning to the hardships of the exhausting work experienced by the healthcare workforce during the first wave of the pandemic.

Second, we verify the stability of the estimates of the baseline specification by including administrative information at the provincial level to capture the objective quality of the

**Table 7. Satisfaction and willingness to change profession or specialization.**

| | All | | | Physicians | | | Nurses | | |
|---|---|---|---|---|---|---|---|---|---|
| | Satisfaction | Prof Change | Spec Change | Satisfaction | Prof change | Spec change | Satisfaction | Prof change | Spec change |
| **Socio-economic factors:** | | | | | | | | | |
| Children | 0.037 | -0.049 | 0.060 | -0.602 | 0.087 | 0.161 | 0.216 | -0.071 | 0.017 |
| | (0.370) | (0.080) | (0.055) | (0.624) | (0.114) | (0.139) | (0.517) | (0.109) | (0.069) |
| Age: > = 30—<40 | -0.688 | 0.388*** | 0.411*** | -0.220 | 0.447** | 0.495*** | -0.678 | 0.331*** | 0.332*** |
| | (0.428) | (0.102) | (0.073) | (1.158) | (0.197) | (0.169) | (0.469) | (0.114) | (0.103) |
| Age: > = 40—<50 | -0.715 | 0.488*** | 0.466*** | -0.797 | 0.605** | 0.470*** | -0.505 | 0.442*** | 0.492*** |
| | (0.570) | (0.153) | (0.079) | (0.981) | (0.259) | (0.152) | (0.574) | (0.166) | (0.106) |
| Age: > = 50—<60 | -0.432 | 0.262* | 0.416*** | -1.671** | 0.501 | 0.299 | 0.250 | 0.182 | 0.483*** |
| | (0.418) | (0.144) | (0.074) | (0.830) | (0.366) | (0.247) | (0.343) | (0.149) | (0.110) |
| Age: > = 60 | 0.433 | -0.050 | 0.265** | 0.000 | 0.268 | 0.137 | 1.594** | -0.414* | 0.352 |
| | (0.800) | (0.187) | (0.119) | (.) | (0.323) | (0.201) | (0.675) | (0.220) | (0.238) |
| Female | 0.382* | -0.201*** | -0.213*** | 0.646 | -0.122 | -0.117 | 0.310 | -0.287*** | -0.334*** |
| | (0.215) | (0.056) | (0.078) | (0.402) | (0.109) | (0.127) | (0.285) | (0.092) | (0.118) |
| Italian | 0.000 | -0.016 | -0.448*** | 0.000 | 0.588 | -0.160 | 0.000 | 0.056 | -0.329** |
| | (.) | (0.193) | (0.171) | (.) | (0.411) | (0.570) | (.) | (0.288) | (0.165) |
| Married | 0.034 | -0.009 | -0.053 | 0.368 | -0.219* | -0.178 | -0.209 | 0.092 | 0.054 |
| | (0.289) | (0.063) | (0.069) | (0.379) | (0.118) | (0.155) | (0.380) | (0.100) | (0.050) |
| House sq. meters >100 | 0.039 | -0.112** | -0.067 | -0.253 | -0.166 | -0.022 | 0.057 | -0.099 | -0.080 |
| | (0.142) | (0.048) | (0.054) | (0.324) | (0.147) | (0.069) | (0.196) | (0.069) | (0.078) |
| Good health status | 0.960*** | -0.459*** | -0.471*** | 1.984*** | -0.880*** | -0.395** | 0.864** | -0.361** | -0.483** |
| | (0.242) | (0.112) | (0.146) | (0.497) | (0.319) | (0.175) | (0.371) | (0.161) | (0.206) |
| Chronic diseases | -0.296* | 0.263*** | 0.177** | -0.703** | 0.323*** | 0.251*** | -0.276 | 0.276*** | 0.176 |
| | (0.167) | (0.059) | (0.071) | (0.321) | (0.094) | (0.054) | (0.280) | (0.095) | (0.109) |
| Living alone | -0.284 | -0.012 | -0.160 | -0.656 | -0.100 | -0.210* | -0.362 | -0.017 | -0.188 |
| | (0.243) | (0.108) | (0.102) | (0.435) | (0.162) | (0.120) | (0.342) | (0.151) | (0.140) |
| Never changed workplace | 0.307 | -0.107*** | -0.176*** | 0.901* | -0.145 | -0.193*** | 0.095 | -0.079 | -0.131* |
| | (0.253) | (0.036) | (0.057) | (0.493) | (0.093) | (0.063) | (0.262) | (0.073) | (0.073) |
| Health workers in the family | -0.059 | 0.065* | 0.046 | 0.493 | 0.183** | 0.055 | -0.109 | -0.018 | 0.056 |
| | (0.173) | (0.039) | (0.052) | (0.422) | (0.077) | (0.083) | (0.267) | (0.070) | (0.071) |
| **Workplace Condition:** | | | | | | | | | |
| Hospital worker | 0.289 | 0.113 | -0.018 | 0.363 | 0.181 | -0.050 | 0.446 | 0.091 | -0.005 |
| | (0.262) | (0.075) | (0.055) | (0.514) | (0.171) | (0.109) | (0.373) | (0.102) | (0.099) |
| Teaching hospital | 0.631* | 0.042 | -0.283*** | 0.000 | -0.031 | -0.559*** | -0.899 | -0.438 | -0.819** |
| | (0.346) | (0.128) | (0.100) | (.) | (0.262) | (0.124) | (0.837) | (0.333) | (0.336) |
| Private | -0.299 | 0.017 | -0.061 | 0.211 | 0.211* | 0.107 | -0.270 | -0.016 | -0.082 |
| | (0.325) | (0.066) | (0.070) | (0.803) | (0.127) | (0.241) | (0.377) | (0.086) | (0.054) |
| Management role | 0.238 | -0.254*** | -0.106 | 0.860 | -0.333*** | -0.385** | -0.286 | -0.236** | 0.143 |
| | (0.239) | (0.086) | (0.079) | (0.584) | (0.127) | (0.167) | (0.429) | (0.095) | (0.100) |
| Contract with work-shifts | -0.240 | -0.002 | 0.101 | -0.178 | 0.237 | 0.223 | -0.413 | -0.060 | 0.134 |
| | (0.231) | (0.121) | (0.080) | (0.309) | (0.197) | (0.177) | (0.475) | (0.115) | (0.100) |
| Average hours worked | -0.029*** | 0.011*** | 0.013*** | -0.052*** | 0.015*** | 0.014** | -0.009 | 0.009* | 0.009** |
| | (0.008) | (0.002) | (0.003) | (0.012) | (0.004) | (0.005) | (0.011) | (0.005) | (0.004) |
| Tenure | -0.007 | 0.003 | -0.001 | 0.029 | -0.001 | 0.010** | -0.016 | 0.000 | -0.009** |
| | (0.014) | (0.003) | (0.002) | (0.025) | (0.004) | (0.005) | (0.016) | (0.005) | (0.003) |
| COVID-19 specialty | -0.274 | 0.036 | -0.089* | 0.827* | 0.112 | 0.231* | -0.479** | -0.031 | -0.245*** |
| | (0.167) | (0.055) | (0.047) | (0.485) | (0.134) | (0.135) | (0.214) | (0.081) | (0.081) |

*(Continued)*

**Table 7.** (Continued)

| | All | | | Physicians | | | Nurses | | |
|---|---|---|---|---|---|---|---|---|---|
| | Satisfaction | Prof Change | Spec Change | Satisfaction | Prof change | Spec change | Satisfaction | Prof change | Spec change |
| High-quality facility | 1.308*** | -0.415*** | -0.500*** | 0.000 | -0.438*** | -0.405*** | 0.732** | -0.430*** | -0.552*** |
| | (0.299) | (0.084) | (0.066) | (.) | (0.091) | (0.081) | (0.296) | (0.109) | (0.084) |
| Lack of medical personnel | -0.586*** | 0.166*** | 0.217** | -0.611 | 0.254* | 0.236** | -0.491*** | 0.117 | 0.217** |
| | (0.198) | (0.062) | (0.085) | (0.442) | (0.131) | (0.108) | (0.187) | (0.085) | (0.091) |
| High salary | 0.405 | -0.266*** | -0.299*** | 0.932* | -0.266 | -0.239 | 1.566** | -0.172 | -0.298* |
| | (0.255) | (0.068) | (0.101) | (0.551) | (0.302) | (0.190) | (0.655) | (0.178) | (0.153) |
| Nurse | -0.076 | 0.159 | 0.073 | | | | | | |
| | (0.333) | (0.108) | (0.127) | | | | | | |
| **COVID-19 specific factors:** | | | | | | | | | |
| COVID-19 Death rate | -0.001 | -0.001 | -0.001*** | -0.002 | 0.001 | 0.002** | 0.001 | -0.002*** | -0.002*** |
| | (0.001) | (0.001) | (0.000) | (0.002) | (0.001) | (0.001) | (0.001) | (0.001) | (0.000) |
| Prompt response | 0.366 | -0.040 | 0.009 | -0.061 | -0.188 | 0.059 | 0.443 | 0.054 | -0.036 |
| | (0.309) | (0.040) | (0.052) | (0.400) | (0.127) | (0.108) | (0.500) | (0.054) | (0.074) |
| Effective response | 0.624** | -0.144** | -0.163*** | 0.303 | 0.051 | -0.063 | 0.855* | -0.301** | -0.218** |
| | (0.316) | (0.071) | (0.043) | (0.350) | (0.141) | (0.108) | (0.439) | (0.124) | (0.109) |
| Infected colleagues | 0.381 | 0.047 | 0.040 | 1.122** | -0.013 | -0.003 | 0.255 | 0.119 | 0.159*** |
| | (0.305) | (0.080) | (0.035) | (0.542) | (0.120) | (0.093) | (0.432) | (0.103) | (0.052) |
| Dead colleagues | -0.357 | 0.124 | -0.095 | -1.468*** | 0.262* | -0.011 | 0.428 | 0.168 | -0.088 |
| | (0.285) | (0.096) | (0.114) | (0.462) | (0.143) | (0.202) | (0.352) | (0.138) | (0.124) |
| COVID-19 overtime | -0.642*** | 0.030 | 0.034 | -0.281 | -0.125 | -0.050 | -0.752** | 0.082 | 0.050 |
| | (0.249) | (0.049) | (0.059) | (0.555) | (0.087) | (0.163) | (0.327) | (0.068) | (0.071) |
| Exposed to COVID-19 | 0.904*** | 0.033 | -0.015 | 0.658 | -0.193 | -0.223 | 1.276*** | 0.065 | 0.037 |
| | (0.347) | (0.081) | (0.095) | (0.733) | (0.130) | (0.140) | (0.373) | (0.126) | (0.130) |
| Positive to COVID-19 | 0.275 | -0.063 | 0.025 | 1.735 | -0.145 | -0.134 | 0.234 | -0.031 | 0.039 |
| | (0.309) | (0.071) | (0.083) | (1.285) | (0.158) | (0.149) | (0.337) | (0.085) | (0.106) |
| Work with COVID-19 positives | 0.028 | -0.151*** | -0.172*** | -0.959* | -0.089 | -0.088 | 0.313 | -0.158** | -0.194*** |
| | (0.248) | (0.040) | (0.039) | (0.564) | (0.133) | (0.122) | (0.251) | (0.069) | (0.050) |
| COVID-19: change of specialty/ function | -0.196 | 0.145* | 0.145** | 0.223 | 0.111 | 0.064 | -0.227 | 0.172* | 0.195*** |
| | (0.153) | (0.075) | (0.064) | (0.400) | (0.138) | (0.110) | (0.223) | (0.097) | (0.068) |
| Constant | 4.468*** | -0.384 | -0.137 | 3.921* | -0.929 | -0.799 | 4.110*** | -0.238 | -0.080 |
| | (0.753) | (0.335) | (0.296) | (2.069) | (0.617) | (0.698) | (1.059) | (0.409) | (0.339) |
| N Obs. | 6,951 | 7,134 | 7,134 | 1,224 | 2,348 | 2,352 | 3,895 | 4,254 | 4,254 |
| Region fixed effect | Yes | Yes | Yes | Yes | Yes | Yes | Yes | Yes | Yes |

OLS regressions. See S2 and S3 Tables for outcome and control definitions, respectively. Standard errors clustered at the level of the region of work in parentheses. Significant at 10% *; significant at 5% **; significant at 1% ***.

healthcare system within which the respondents operate. There are four measures of objective quality. The first measure proxies for workforce availability and coincides with the rate of physicians registered with the provincial board of physicians per 10,000 inhabitants (a correlation of -0.063 with the perceived lack of personnel). The other three measures are the 30-day readmission rate for acute myocardial infarction (AMI), the 30-day readmission rate for stroke and the 30-day readmission rate for chronic obstructive pulmonary disease (COPD) in 2019 as measured by the Ministry of Health in the "National Healthcare Outcomes Program" (correlations of -0.042, -0.091, and -0.073 with perceived quality, respectively). These 30-day

readmission rates are computed as the ratio between the number of readmissions for the related disease within 30 days of discharge out of the total number of admissions due to the given disease (e.g., the number of readmissions due to stroke out of the overall admissions due to stroke). Note that the inclusion of administrative proxies for objective quality does not affect the estimates of the self-perceived quality measures (S5 Table), highlighting the relevance of self-perception over objective factors.

Third, we consider alternative definitions of *Satisfaction*. Specifically, we work with its discrete version, *Satisfaction 2*, which ranges from 8 to 40, being the sum of the 8 categorical variables related to *Profession*, *Job*, *Salary*, *Work-life balance*, *Relationship with the colleagues*, *Relationships with the administration*, *Work hours*, and *Career*. Each of these variables is measured on a 5-item Likert scale ranging from 1 (very dissatisfied) to 5 (very satisfied). Alternatively, *Satisfaction 3* was computed as the arithmetic mean of the same 8 categorical variables on which we also performed a principal component analysis (PCA) obtaining *Satisfaction PCA*; that is, a continuous outcome varying from -5.42 to 4.69 (S2 Table). As reported in S6 Table, the coding of *Satisfaction* into a binary outcome does not drive our results, as we find no significant difference in the explanatory power of each control group (i.e., personal, contextual and COVID-19-related factors).

## Discussion

Our results show how contextual factors explain a remarkable amount of the variation in the level of satisfaction (58%), the willingness to change profession (43%) and the willingness to change specialization (52%), while personal factors, however important, matter to a lesser extent (21% of satisfaction, 49% of changing profession, and 39% of changing specialization). In particular, working for a (perceived) high-quality facility and in a province where the worker perceives a lack of medical personnel are the two main components of job satisfaction and of the willingness to change profession or specialization. These findings hold for different types of professionals (e.g., nurses and physicians) and are robust to several checks.

Our findings reinforce previous evidence, particularly on the relevance of contextual factors related to the workplace in determining job satisfaction. Numerous studies highlight that, in non-emergency times, healthcare workers' satisfaction is significantly and negatively associated with workload and working shifts [37–41] and access to resources [42, 43] but positively associated with economic incentives [44], quality [40–42, 45–48] and having a managerial role and coordination responsibilities [49, 50]. Similarly, these factors also matter during the pandemic. [20] find job satisfaction among Jordanian physicians to be positively associated with age and salaries and negatively associated with working as a general practitioner, as a specialist or in high-load hospitals. The study by [21] shows that the number of office days is an important determinant of job satisfaction and that turnover intentions vary with age among healthcare workers in Bolivia. Similarly, age and workload appear to be the main drivers of job satisfaction and of the willingness to leave the profession among Egyptian nurses [22]; Spanish nurses' job satisfaction is primarily affected by their workload, access to resources and information [23].

However, in contrast to what is commonly expected, the intensity of the spread of the pandemic did not significantly affect workers' satisfaction or undermine their vocation, either when it is captured by the administrative measure (i.e., the death rate at the provincial level), or by their personal experience with the virus (e.g., being infected). Rather, job satisfaction is significantly reduced by working overtime and by dealing with infected patients; that is, by changes in the working conditions caused by the health emergency.

Moreover, when we compare the determinants of the propensity to change profession or specialization, nurses' resilience stands out. In particular, nurses who worked in a COVID-19 related ward are significantly less willing to change specialization, and nurses who had direct contact with COVID-19 patients are less willing to change both profession and specialization. However, if nurses had infected colleagues or had been reassigned to a different ward or function, they are more likely to consider a change. Among physicians, none of the COVID-19-related factors influenced their vocation in terms of either profession or specialization. The intensity of the outbreak of the pandemic mainly affected nurses, although the effects are quite small in magnitude with respect to the other controls.

In our sample, approximately 50% of physician respondents and 79% of nurse respondents are female, in line with female participation in the public healthcare sector (i.e., 48% of physicians and 78% of nurses) [51]. Regarding age, our sample is also consistent with the national trend of physicians being, on average, older than nurses (49 vs. 40) [32]. Hence, the present study further contributes to the literature by increasing the representativeness of the recruited sample with respect to the general population of healthcare workers.

Overall, our results have important policy implications. On the one hand, they highlight the commitment of healthcare workers whose vocation is not challenged by their own or their colleagues' struggle with the virus. On the other hand, they indicate that factors that can be affected by policy interventions, such as those related to the workplace, are the main drivers of job satisfaction. Health emergencies, such as the COVID-19 outbreak, undermine workers' commitment, not necessarily out of fear of contracting the virus but by worsening the working environment. Indeed, job satisfaction and commitment are preserved mainly by guaranteeing good conditions in the working environment, both in normal times and during an emergency. In particular, it appears crucial to foster the quality of facilities and to reduce shortages of medical personnel, since these interventions would improve the provision of care to patients while simultaneously supporting healthcare workforce satisfaction. For a one-standard-deviation increase in healthcare facility quality, the level of job satisfaction increases by 0.40 p.p., which is equal to 7% of the mean value; however, a one-standard-deviation increase in the perceived lack of medical personnel decreases job satisfaction by 0.06 p.p., which is equal to 1.1% of the mean.

## Limitations

This study has several limitations. First, our analysis uses a cross-sectional approach rather than a longitudinal perspective. While a panel study would have been ideal to evaluate the change in the level of satisfaction with respect to the period before the pandemic, our approach provides a valid estimation of how job satisfaction varies with both personal and contextual elements and with COVID-19-related factors. In particular, since most of the personal and contextual factors do not vary over time, especially in a short time period (e.g., one year), one can reasonably expect our analysis of the impact of these factors on job satisfaction to be well identified. This is also supported by the fact that our results are well in line with the findings reported in the literature of interest. Moreover, participants were asked to describe the working conditions considering solely the period from the end of February 2020 to the beginning of June 2020. Therefore, the inclusion of the COVID-19-related factors in the regressions allows us to capture the relationship between the working conditions during the emergency situation (e.g., working overtime) and the level of job satisfaction, in addition to the standard working conditions (e.g., average hours worked, or contract with work shifts). However, future studies on this topic seeking to account for the effects of time-varying factors should use a longitudinal approach instead.

Second, participants voluntarily participated in our survey; thus, they may be self-selected, and reporting bias cannot be excluded. For instance, if individuals with low job satisfaction were more likely to respond (e.g., to express dissatisfaction), then the estimated satisfaction level may be lower than the worker population average. By contrast, if individuals with below-average satisfaction were less likely to respond (e.g., due to distrust in the system), then the estimated satisfaction level may be higher than the worker population average. Reassuringly, however, our sample is representative of the underlying worker population by gender and age.

Third, it is not possible to compute an exact response rate since our survey circulated also through the provincial boards of physicians and nurses, hospital websites, and representative associations. However, this decentralized approach allowed us to target all Italian healthcare workers rather those working in specific hospitals or geographical areas within a country, as most previous studies have done.

## Conclusion

Immediately after the first wave of the COVID-19 pandemic, we conducted a unique survey of Italian healthcare workers to explore the determinants of their professional satisfaction and vocation focusing on personal, contextual and COVID-19-related factors.

In addition to confirming the role of gender, age, good health and chronic diseases among the personal factors, the analysis shows that contextual factors are the strongest determinants of workers' satisfaction and propensity to change profession or medical specialization. In particular, we find that *working in a high-quality facility* has beneficial effects on workers, increasing work-related satisfaction and willingness to remain in the profession and in the medical specialization. However, *working in a province with a serious shortage of medical personnel* yields the opposite result. Our findings have strong policy implications because the main drivers of professional satisfaction are modifiable. Hence, policymakers should implement effective strategies to improve working conditions in the healthcare sector in general and further support workers along these dimensions in emergency times. This would directly impact the turnover of healthcare workers while indirectly increasing the quality of care for patients. Although our analysis does not offer a one-size-fits-all policy to improve working conditions in the healthcare sector, in the specific case of Italy, policymakers should foster quality of facilities and invest in increasing the number of medical personnel.

In examining the intensity of COVID-19 exposure, we find that work accidents, such as being infected or losing colleagues to the virus, do not play a relevant role in affecting the vocation of healthcare workers. Rather, we find they are more affected by changes in working conditions caused by the pandemic, such as having to work overtime or being reassigned to a different ward/function. Healthcare professionals are devoted to helping others, and supporting them through difficult times such that a severe pandemic in the province of work plays a marginal role; more important, it contradicts the common expectation. Indeed, healthcare workers and, especially, nurses are even more satisfied with their job and less prone to change profession or specialization in the most affected provinces following the first wave of the pandemic, further showing the resilience of their vocation.

## Supporting information

**S1 Fig. Severity of the COVID-19 first-wave by region.** The *Death Rates* (A) and *Share of COVID-19 Deaths* (B) are measures computed by the National Institute of Statistics (Istat) together with the Istituto Superiore di Sanitá (Iss) on administrative data [33]. The index *Death Rates*, referring to the period January-May 2020, represents the mortality rate due to COVID-19 standardized by the demographic characteristics of the resident population in each

province (values expressed per 100,000 inhabitants). The *Share of COVID-19 Deaths* is the proportion of deaths by COVID-19 cases over the total number of deaths in the relevant time and location. The shapefile used to create the map is plotting the boundaries of the regional administrative units as of January 2020 and it is retrieved from the National Institute of Statistics (Istat). Elaboration by the authors.
(PDF)

**S2 Fig. Survey participation.** The graph shows the cumulative number of responses to the online survey by day of participation (i.e., day when the completed survey was submitted and the response was registered by the Google Form platform). The vertical line identifies the end of the pilot run during the first week (June 15*th* to June 22, 2020) to verify the clarity of the questionnaire. No issue arose during the pilot, and therefore we proceeded using all responses collected in the analysis.
(PDF)

**S3 Fig. Timeline of the survey.**
(PDF)

**S4 Fig. Survey participation by region.** The maps present the absolute numbers of participants by region and professional category, i.e., all healthcare workers (A), physicians (B), and nurses (C). Data reported by quintiles of the professional category considered. The shapefile used to create the map is plotting the boundaries of the regional administrative units as of January 2020 and it is retrieved from the National Institute of Statistics (Istat). Elaboration by the authors.
(PDF)

**S1 Table. Associations participating in the survey.** Professional associations that shared our survey with their members through their website and/or their mailing list.
(PDF)

**S2 Table. Outcome definitions.** Definition of the outcomes of interest, alternative outcomes and individual components of satisfaction. Individual components are defined as dummies, moving from categorical variables originally measured on a 5-item scale.
(PDF)

**S3 Table. Variables definition.** When we refer to the COVID-19 crisis, we refer to the first wave that took place in Italy from the end of February 2020 to the beginning of June 2020. AMI = Acute myocardial infraction. COPD = Chronic obstructive pulmonary disease. All 30-day readmission rates have been collected from the 'National Healthcare Outcomes Program" ("Piano Nazionale Esiti—PNE), which is a national program run since 2012 by the Ministry of Health intended to develop and implement practical indexes to measure, analyze, evaluate and monitor the performance of healthcare facilities operating within the Italian healthcare system.
(PDF)

**S4 Table. Satisfaction and willingness to change profession or specialization—robustness of COVID-19 factors.** OLS regressions. See S2 and S3 Tables for outcome and control definitions, respectively. Standard errors clustered at the level of the region of work in parentheses. Significant at 10% *; significant at 5% **; significant at 1% ***.
(PDF)

**S5 Table. Satisfaction and willingness to change profession or specialization—robustness of administrative information.** OLS regressions. See S2 and S3 Tables for outcome and

control definitions, respectively. Standard errors clustered at the level of the region of work in parentheses. Significant at 10% *; significant at 5% **; significant at 1% ***. *Physicians/10.000 inhabitants* administrative information reporting the number of physicians registered in each province weighted by the provincial resident population (source: Health-for-All Italy. Year 2019). *PNE 30days readmissions* are three standardized measures reporting the provincial rate of readmission to hospital 30 days after discharge for selected diseases (i.e., acute myocardial infarction—AMI, stroke, chronic obstructive pulmonary Disease—COPD) (source: Piano Nazionale Esiti-PNE- Ministry of Health. Year: 2019).
(PDF)

**S6 Table. Satisfaction—robustness of alternative outcomes.** OLS regressions. See S2 and S3 Tables for outcome and control definitions, respectively. Standard errors clustered at the level of the region of work in parentheses. Significant at 10% *; significant at 5% **; significant at 1% ***.
(PDF)

# Acknowledgments

We are grateful for the support of the physicians and nurses associations that promoted the dissemination of our survey, and we thank all the physicians, nurses, biologists, psychologists, obstetricians, and technicians who took the time to complete it.

## Author Contributions

**Conceptualization:** Paola Bertoli, Veronica Grembi.

**Data curation:** Paola Bertoli, Veronica Rattini.

**Formal analysis:** Emilia Barili.

**Investigation:** Paola Bertoli, Veronica Rattini.

**Supervision:** Veronica Grembi.

**Visualization:** Emilia Barili, Veronica Rattini.

**Writing – original draft:** Paola Bertoli.

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
