## [Editor Report · Decision Letter 0]

31 Jan 2022

PONE-D-22-00138Job Satisfaction Among Healthcare Workers in the Aftermath of the COVID-19 Pandemic1PLOS ONE

Dear Dr. Grembi,

Thank you for submitting your manuscript to PLOS ONE. After careful consideration, we feel that it requires modifications to meet PLOS ONE’s publication criteria as it currently stands. Therefore, we invite you to submit a revised version of the manuscript that addresses the points raised during the review process.

 Prior to further steps, it is necessary to organize your manuscript per the PLOS ONE's guidelines (https://journals.plos.org/plosone/s/submission-guidelines):

We look forward to receiving your revised manuscript.

Kind regards,

Farzad Taghizadeh-Hesary

Academic Editor

PLOS ONE

Journal Requirements:

3. During the internal evaluation of the submission we have noted the following statement: "f this research was approved by the Ethics committee of

the University of Pavia (Italy) to which Emilia Barili affiliated at the time of the survey" Please could you clarify whether author Emilia Barili is affiliated with the research ethics committee which provided ethical approval. If no, please could you revise this text to avoid confusion.

4. Please update your submission to use the PLOS LaTeX template. The template and more information on our requirements for LaTeX submissions can be found at http://journals.plos.org/plosone/s/latex.

"Paola Bertoli is the recipient of a Rita Levi Montalcini Fellowship to promote the moving back in Italian University of Young Italian Scholars based abroad and willing to come back to Italy"

7. We note that Figures 1 and 3 in your submission contain copyrighted images. All PLOS content is published under the Creative Commons Attribution License (CC BY 4.0), which means that the manuscript, images, and Supporting Information files will be freely available online, and any third party is permitted to access, download, copy, distribute, and use these materials in any way, even commercially, with proper attribution. For more information, see our copyright guidelines: http://journals.plos.org/plosone/s/licenses-and-copyright.

a. You may seek permission from the original copyright holder of Figures 1 and 3 to publish the content specifically under the CC BY 4.0 license. 

Additional Editor Comments:

Thank you for submitting the results of your study to the PLOS ONE.

Prior to further steps, it is necessary to organize your manuscript per the PLOS ONE's guidelines (https://journals.plos.org/plosone/s/submission-guidelines):

Manuscripts should be organized as follows:

Beginning section

The following elements are required, in order:

Title page: List title, authors, and affiliations as first page of the manuscript

Abstract

Introduction

Middle section

The following elements can be renamed as needed and presented in any order:

Materials and Methods

Results

Discussion

Conclusions (optional)

Ending section

The following elements are required, in order:

Acknowledgments

References

Supporting information captions (if applicable)

Other elements

Figure captions are inserted immediately after the first paragraph in which the figure is cited. Figure files are uploaded separately.

Tables are inserted immediately after the first paragraph in which they are cited.

Supporting information files are uploaded separately.
---

## [Author Response · Author response to Decision Letter 0]

16 May 2022

Dear Prof. Taghizadeh-Hesary,

Thank you for your letter of January 31st. We reviewed the paper as to make sure it complies with PLOS ONE's style requirements and we adopted the PLOS ONE’s Latex template. Moreover, we now clearly specify in the text that each participant provided written informed consent that was embedded in the first page of the questionnaire and we provide the specific details on the Ethics committee that approved our survey (i.e., the Ethics committee of the University of Verona, called Comitato di Approvazione per la Ricerca sull'Uomo'').

We also confirm that the reported funder had no role in study design, data collection and analysis, decision to publish, or preparation of the manuscript.

In what follows, we answer to each of the specific points raised in your letter detailing how these have been taken into account in the revised version. We hope that this revised version will bring us nearer to a publication in PLON ONE and we again thank you for this opportunity and look forward to hearing from you at your convenience.

Best regards,

Emilia Barili, Paola Bertoli, Veronica Grembi and Veronica Rattini

 

COMMENTS AND RELATED REPLIES

A: The manuscript has been reviewed to ensure compliance with PLOS ONE’s style requirements. Specifically, we adapt the Title formatting and insert the correct authors’ information. We eliminate the Section and Subsection numbering. The main sections are now Introduction, Materials and Methods, Results and Discussion, Conclusion. We took care of Tables and Figures formatting the related captions, as well as of footnotes.

A: We obtained written informed consent from each participant at the beginning of the questionnaire. Specifically, the first screen provided participants with information about the content of the survey, its intended use and a privacy statement. After this information, each participant was asked whether she agreed to provide her consent for participation. If participants clicked on the option “Yes, I consent to participate in the survey”, they proceeded with the questionnaire. If they opted for “No, I do not consent to participate in the survey”, they were re-directed to an ending screen. Our study does not include minors and does not use medical records or achieved samples. 

3. During the internal evaluation of the submission we have noted the following statement: "this research was approved by the Ethics committee of the University of Pavia (Italy) to which Emilia Barili affiliated at the time of the survey" Please could you clarify whether author Emilia Barili is affiliated with the research ethics committee which provided ethical approval. If no, please could you revise this text to avoid confusion.

A: Emilia Barili was affiliated with the University of Pavia at the time of the survey, but she was not directly affiliated or connected to the Ethics committee of the University of Pavia

4. Please update your submission to use the PLOS LaTeX template. The template and more information on our requirements for LaTeX submissions can be found at http://journals.plos.org/plosone/s/latex.

A: We adopted the PLOS LaTex template as requested.

"Paola Bertoli is the recipient of a Rita Levi Montalcini Fellowship to promote the moving back in Italian University of Young Italian Scholars based abroad and willing to come back to Italy"

A: The funder had no role in study design, data collection and analysis, decision to publish, or preparation of the manuscript. 

A: We now specify in the “Materials and Methods” section what follows: “The survey was approved by the ``Comitato di Approvazione per la Ricerca sull'Uomo'', that is, the Ethics Committee of the University of Verona (Prot. N. 0221872 - 22/06/2020) and registered at AEA-RCT registry (AEARCTR-0007419), while the University of Pavia certified compliance with privacy requirements (Prot. N. 61080 - 15/06/2020). All participants gave their written informed consent that was embedded on the first page of the questionnaire.” 

7. We note that Figures 1 and 3 in your submission contain copyrighted images. All PLOS content is published under the Creative Commons Attribution License (CC BY 4.0), which means that the manuscript, images, and Supporting Information files will be freely available online, and any third party is permitted to access, download, copy, distribute, and use these materials in any way, even commercially, with proper attribution. For more information, see our copyright guidelines: http://journals.plos.org/plosone/s/licenses-and-copyright.

a. You may seek permission from the original copyright holder of Figures 1 and 3 to publish the content specifically under the CC BY 4.0 license. 

A: Figures 1 and 3 do not contain copyrighted images, but rather images we produced. We now specify this in the footnotes to the figures.

Additional Editor Comments:

Thank you for submitting the results of your study to the PLOS ONE.

Prior to further steps, it is necessary to organize your manuscript per the PLOS ONE's guidelines (https://journals.plos.org/plosone/s/submission-guidelines):

Manuscripts should be organized as follows:

Beginning section

The following elements are required, in order:

Title page: List title, authors, and affiliations as first page of the manuscript

Abstract

Introduction

Middle section

The following elements can be renamed as needed and presented in any order:

Materials and Methods

Results

Discussion

Conclusions (optional)

Ending section

The following elements are required, in order:

Acknowledgments

References

Supporting information captions (if applicable)

Other elements

Figure captions are inserted immediately after the first paragraph in which the figure is cited. Figure files are uploaded separately.

Tables are inserted immediately after the first paragraph in which they are cited.

Supporting information files are uploaded separately.

A: We re-organized the structure of the paper as requested. In particular, the paper now includes Abstract, Introduction, a section providing information on the institutional setting, Materials and Methods, Results and Discussion, Conclusions. We also included the applicable required elements as, among others, acknowledgements and references.

---

## [Decision Letter · Decision Letter 1]

20 Jun 2022

PONE-D-22-00138R1Job Satisfaction Among Healthcare Workers in the Aftermath of the COVID-19 Pandemic1PLOS ONE

Dear Dr. Grembi,

Thank you for submitting your manuscript to PLOS ONE. After careful consideration, we feel that it has merit but does not fully meet PLOS ONE’s publication criteria as it currently stands. Therefore, we invite you to submit a revised version of the manuscript that addresses the points raised during the review process.

ACADEMIC EDITOR: Please find the reviewers' comments to improve the manuscript.

We look forward to receiving your revised manuscript.

Kind regards,

Farzad Taghizadeh-Hesary

Academic Editor

PLOS ONE

Reviewers' comments:

Reviewer's Responses to Questions

**Comments to the Author**

1. If the authors have adequately addressed your comments raised in a previous round of review and you feel that this manuscript is now acceptable for publication, you may indicate that here to bypass the “Comments to the Author” section, enter your conflict of interest statement in the “Confidential to Editor” section, and submit your "Accept" recommendation.

Reviewer #1: All comments have been addressed

Reviewer #2: All comments have been addressed

Reviewer #3: (No Response)

Reviewer #4: All comments have been addressed

2. Is the manuscript technically sound, and do the data support the conclusions?

Reviewer #1: Yes

Reviewer #2: Yes

Reviewer #3: Partly

Reviewer #4: Yes

3. Has the statistical analysis been performed appropriately and rigorously? 

Reviewer #1: Yes

Reviewer #2: Yes

Reviewer #3: Yes

Reviewer #4: Yes

4. Have the authors made all data underlying the findings in their manuscript fully available?

Reviewer #1: Yes

Reviewer #2: Yes

Reviewer #3: Yes

Reviewer #4: Yes

5. Is the manuscript presented in an intelligible fashion and written in standard English?

Reviewer #1: Yes

Reviewer #2: Yes

Reviewer #3: No

Reviewer #4: Yes

6. Review Comments to the Author

Reviewer #1: Manuscript presents a current and important theme in the elaboration of human resources policies in the health area, the COVID-19 pandemic was a learning experience, including the need for rapid activation of emergency programs in public health. It presents important and significant results. I add some considerations for the authors' evaluation.

1- How were the other health professionals (571) included in the analysis? Or just in the methodology of the study?

2- The authors mention the nurses' resilience and that they have less intention of changing profession or specialization. Resilience is due only to personal factors or also to the employability of nurses compared to doctors, reducing the intention to change profession or specialization.

Reviewer #2: #P 2 Line 37: after [18] There must be some word, it seems to be missing.

#P2 Line 46-53 : These statements should not be included in the introduction part with the discussion, theses should be moved to the discussion part. As well as line 64 onwards till the end of the introduction part.

Introduction part should be written in past tense as well the methodology part.

#P5 Research design should the written before the heading of "materials and methods"

Reviewer #3: Dear author, this is an interesting paper about the job satisfaction of healtcare workers after the onset of Covid-19 pandemic. However, the study design doesn't allow to evaluate it's effective impact on those workers, because no data are available about the status of the sample before the pandemic. A longitudinal study could be the best choice to perform the evaluation you intented to do. In addition, to make a stronger evaluation of job satisfaction, you could use a validated tool like The Job Satisfaction Survey (JSS) or others. Other issues with this manuscript are the follow:

- Many studies performed in italy in the same period have not been cited, even if they could help to better understand the context in which the study has been performed (i.e. De Sio et al).

- English language is quite good even if there are many long sentences and others are quite complex to be understood.

- No response rate has been provided.

- The introduction section is too long and, at the end, it deals with results and conclusions that should be put at the end of the paper, not at the beginning.

- The limits of the research have not been disclosed (selection bias etc)

- The authors state that they focused on the respondents from northern Italy, but they didn't indicate how many respondents came from there and how many from other parts of the country

- Many question of the survey have not been justified in relation to the aims of the study (i.e. homes dimensions etc).

I think that the paper needs those and more others corrections to be accepted for publication.

Reviewer #4: 1. Introduction: "with more than 7,000 respondents (about 2,500 physicians and 4,500 nurses) conducted". Please designate the exact numbers of participants.

2. Introduction section is too long. It is recommended the authors summarize the Introduction section (in 1000-1500 words) representing the importance of the study and how the study can fill the literature gap. Many paragraphs-describing the similar articles in other countries- can be discussed in the Discussion section.

3. Please follow the guidelines of scientific writing.

4. Please cite the following Covid-19 related article in the Introduction section:

- https://www.ncbi.nlm.nih.gov/pmc/articles/PMC8184167/

5. Introduction: It may not be necessary to describe the first Covid-19 wave in Italy in details. It is suggested the authors summarize this section in a paragraph.

6. Figs. 1 and 3 can be submitted as supplementary materials.

7. Please follow the Journal guideline to prepare the manuscript. For example, the Results and Discussion sections must be presented separately. It is recommended the author refer to the PLOS ONE guideline for authors.

8. Please add a table summarizing the basic characteristics of the participants.

9. Many figures have been submitted. It is recommended the authors merge Figs. 4-12 to three figures. Figure 13 can be explained in the text and be omitted. Figure 2 can be submitted as a supplementary material.

7. PLOS authors have the option to publish the peer review history of their article (what does this mean?). If published, this will include your full peer review and any attached files.

Reviewer #1: **Yes: **Maria Carlota Borba Brum

Reviewer #2: No

Reviewer #3: No

Reviewer #4: No

---

## [Decision Letter · Decision Letter 2]

14 Sep 2022

Job Satisfaction Among Healthcare Workers in the Aftermath of the COVID-19 Pandemic1

PONE-D-22-00138R2

Dear Dr. Grembi,

We’re pleased to inform you that your manuscript has been judged scientifically suitable for publication and will be formally accepted for publication once it meets all outstanding technical requirements.

Kind regards,

Farzad Taghizadeh-Hesary

Academic Editor

PLOS ONE

Additional Editor Comments (optional):

Reviewers' comments:

Reviewer's Responses to Questions

**Comments to the Author**

1. If the authors have adequately addressed your comments raised in a previous round of review and you feel that this manuscript is now acceptable for publication, you may indicate that here to bypass the “Comments to the Author” section, enter your conflict of interest statement in the “Confidential to Editor” section, and submit your "Accept" recommendation.

Reviewer #3: All comments have been addressed

2. Is the manuscript technically sound, and do the data support the conclusions?

Reviewer #3: Yes

3. Has the statistical analysis been performed appropriately and rigorously? 

Reviewer #3: Yes

4. Have the authors made all data underlying the findings in their manuscript fully available?

Reviewer #3: Yes

5. Is the manuscript presented in an intelligible fashion and written in standard English?

Reviewer #3: Yes

6. Review Comments to the Author

Reviewer #3: All my questions have been addressed. I think that the paper has improved and it is now suitable for puplication.

7. PLOS authors have the option to publish the peer review history of their article (what does this mean?). If published, this will include your full peer review and any attached files.

Reviewer #3: **Yes: **GIUSEPPE BUOMPRISCO

---

## [Editor Report · Acceptance letter]

28 Sep 2022

PONE-D-22-00138R2 

Job satisfaction among healthcare workers in the aftermath of the COVID-19 pandemic 

Dear Dr. Grembi:

I'm pleased to inform you that your manuscript has been deemed suitable for publication in PLOS ONE. Congratulations! Your manuscript is now with our production department. 

Kind regards, 

on behalf of

Dr. Farzad Taghizadeh-Hesary 

Academic Editor

PLOS ONE